# An Efficient and Secure Certificateless Aggregate Signature Scheme for Vehicular Ad hoc Networks

Asad Iqbal [1], Muhammad Zubair [1], Muhammad Asghar Khan [2], Insaf Ullah [2], Ghani Ur-Rehman [1], Alexey V. Shvetsov [3,4,*] and Fazal Noor [5,*]

1   Department of Computer Science, Khushal Khan Khattak University, Karak 27200, Pakistan; asadiqbal628@gmail.com (A.I.); dr.muhammadzubair@kkkuk.edu.pk (M.Z.); ghani.rehman@kkkuk.edu.pk (G.U.-R.)
2   Faculty of Engineering Sciences and Technology, Hamdard University, Islamabad 44000, Pakistan; m.asghar@hamdard.edu.pk (M.A.K.); insaf.ullah@hamdard.edu.pk (I.U.)
3   Department of Smart Technologies, Moscow Polytechnic University, St. Bolshaya Semenovskaya, 38, 107023 Moscow, Russia
4   Department of Car Transport Operation and Car Service, North-Eastern Federal University, St. Belinsky, 58, 677000 Yakutsk, Russia
5   Department of Computer and Information Systems, Islamic University of Madinah, Madinah 400411, Saudi Arabia
*   Correspondence: a.shvetsov@vvsu.ru (A.V.S.); mfnoor@iu.edu.sa (F.N.)

**Abstract:** Vehicular ad hoc networks (VANETs) have become an essential part of the intelligent transportation system because they provide secure communication among vehicles, enhance vehicle safety, and improve the driving experience. However, due to the openness and vulnerability of wireless networks, the participating vehicles in a VANET system are prone to a variety of cyberattacks. To secure the privacy of vehicles and assure the authenticity, integrity, and nonrepudiation of messages, numerous signature schemes have been employed in the literature on VANETs. The majority of these solutions, however, are either not fully secured or entail high computational costs. To address the above issues and to enable secure communication between the vehicle and the roadside unit (RSU), we propose a certificateless aggregate signature (CLAS) scheme based on hyperelliptic curve cryptography (HECC). This scheme enables participating vehicles to share their identities with trusted authorities via an open wireless channel without revealing their identities to unauthorized participants. Another advantage of this approach is its capacity to release the partial private key to participating devices via an open wireless channel while keeping its identity secret from any other third parties. A provable security analysis through the random oracle model (ROM), which relies on the hyperelliptic curve discrete logarithm problem, is performed, and we have proven that the proposed scheme is unforgeable against Type 1 ($FGR_1$) and Type 2 ($FGR_2$) forgers. The proposed scheme is compared with relevant schemes in terms of computational cost and communication overhead, and the results demonstrate that the proposed scheme is more efficient than the existing schemes in maintaining high-security levels.

**Keywords:** vehicular ad hoc network; security; certificateless aggregate signature; hyperelliptic curve cryptography

## 1. Introduction

Vehicular ad hoc network (VANET) is an advanced version of the mobile ad hoc network (MANET), which was developed to enhance the safety, efficiency, and convenience of transportation [1]. VANET is a set of applications designed to offer new services connected to the traffic management system, designed to help various users to be better informed and use the transportation network to be safer, more connected, and significantly more intelligent. VANET can be established for monitoring and controlling traffic using the concept

of vehicle-to-everything (V2X) communication, which includes vehicle to infrastructure (V2I), vehicle to sensor (V2S), vehicle to pedestrian (V2P), and Vehicle to Vehicle (V2V) communication [2,3]. The general architecture of a VANET is shown in Figure 1, which includes vehicles with built-in 5G-enabled onboard units (OBUs), 5G-enabled roadside units (RSUs), and trusted authorities (TAs). TA is a service provider, ensuring the safety of the VANET network and generating public and private keys for OBUs and RSUs [4]. In VANETs, each vehicle communicates through an OBU and broadcasts traffic-related information such as positions, speed, current time, traffic and road conditions to a nearby vehicle and an RSU [5].

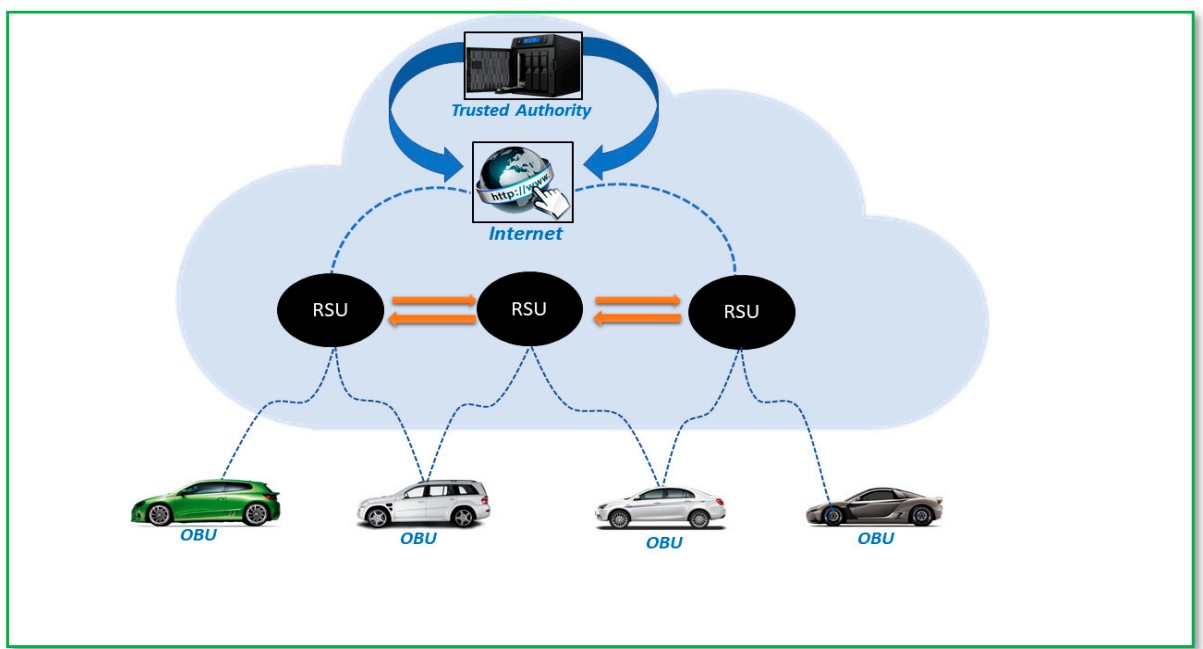

**Figure 1.** The general architecture of a VANET.

Despite all the attractive features offered by a VANET, there are significant challenges regarding its security and privacy when information is shared among vehicles through an open wireless channel. In a VANET system, an attacker can send bogus messages to the RSUs or other OBUs, which might cause disturbance on the roads, so it is necessary to verify the authenticity and integrity of the message [6]. Digital signature-based authentication techniques for VANETs have been built in various cryptographic frameworks, including public-key infrastructure (PKI), identity-based (ID), and certificateless cryptography.

In the standard implementation of public-key cryptography (PKC), each public key is required to produce a corresponding digital certificate [7]. Nevertheless, it not only involves certificate management but also contributes to an increase in the verification cost. To address the issues of certificate management, Shamir [8] came up with a novel technique in 1984 called ID-based public-key cryptography (ID-PKC). The public key consists of the user's identity, such as phone number, e-mail address, etc., eliminating the need for a certificate. However, the private key is generated by the Key Generation Center (KGC), which might lead to a key escrow problem. To overcome the shortcomings of key management in traditional PKC and the key escrow problem in ID-PKC, Al-Riyami and Paterson [9] proposed Certificateless Public Key Cryptography (CL-PKC), which also requires the KGC to generate part of the user's private key.

In contrast, the other part is generated by the user locally. As a result, key escrow is overcome, and CL-PKC does not need to create certificates. Many researchers combine certificateless cryptography with aggregate signature (AS) to create certificateless aggregate signature (CLAS) schemes, which minimize node authentication overhead and address certificate management and key escrow issues in classical cryptosystems. CLAS can prevent

the routing information from being forged, altered, or impersonated, as well as ensure its integrity and provide authentication and nonrepudiation for the sender.

Some entities in VANETs, such as RSUs and OBUs, have limited computing and storage capacity; therefore, efficiency must be taken into account while designing a suitable authentication system for efficient communication in VANETs. In 2003, Boneh et al. [10] proposed the idea of an aggregate signature, which combines the signatures of multiple messages sent by various vehicles into a single, short signature. By using aggregate signatures, we can reduce computational and communicational costs and increase storage capacity to enhance the efficiency of a VANET system.

In this paper, we propose a certificateless aggregate signature scheme based on hyperelliptic curve cryptography (HECC), which is the advanced version of Elliptic Curve Cryptography (ECC), using a key size of 80 bits, providing the same security benefits of ECC, but with less computational cost, communication overhead, and memory requirement. HECs are algebraic curves with the genus $\bar{g} \geq 1$; since the field of an HECC is a quadratic extension of the field of rational functions, we can say that it is the simplest field of algebraic functions, except the field of rational functions [11]. HECC consists of the divisor $\mathcal{D}_P$, the finite formal sum of points on a hyperelliptic curve. The divisor $\mathcal{D}_P$ also forms an Abelian group known as the Jacobian group [12]. Based on the above discussion, this research article contributes to the security of VANETs through the following key characteristics:

- In this article, we propose an efficient certificateless aggregate signature scheme for the security and privacy protection of VANETs using hyperelliptic curve cryptography;
- The proposed scheme enables participating vehicles to share their identities with trusted authorities via an open channel without revealing their identities to unauthorized participants; as a result, sender and recipient anonymity will be ensured;
- In addition, this scheme will disclose the partial private key to participating devices via an open channel while keeping it concealed from other third parties;
- Finally, the noteworthy feature of the proposed scheme is its utilization of a hyperelliptic curve to generate and verify signatures with less computational and communication costs.

The subsequent sections of this article are organized in the following manner: Section 2 discusses the literature review. The proposed scheme is described in Section 4, while Section 3 discusses the necessary preliminary steps. Section 5 deals with the security evaluation. An analysis of performance is covered in Section 6. Concluding remark on the proposed scheme is detailed in Section 7.

## 2. Literature Review

A VANET is a communication technology that enables V2V and V2I communications via the Internet, which can be affected by several cyber-attacks. So, to avoid such circumstances, the best solution is authentication, in which the participating nodes in the VANET environment can authenticate each other before transferring data or information. To achieve authentication, the best approach is to use a digital signature, which allows a sender to sign data with his private key and deliver it to the recipient, who can then use the sender's public key to verify the signature.

In a typical PKC-based signature, each user needs to produce a valid digital certificate that contains information about the identity of the certificate owner and the public key [7]. However, it not only requires certificate management but also contributes to an increase in the verification cost. To address certificate management issues, Shamir [8] proposed ID-PKC, in which the user's identity is his public key, bypassing the need for certificates. However, the private keys are generated via the KGC, which might lead to a key escrow problem.

Al-Riyami and Paterson [9] proposed the idea of CL-PKC, in which the user's private key is made up of a secret value and a partial private key. The user chooses his secret key, while the KGC generates the partial private key. Since the KGC cannot access the user's complete private key without the user's secret key, the user's public key can be calculated

from the secret value. Thus, the potential security issues associated with key escrow are eliminated. Secondly, the user's public key can be calculated from the secret value, so a certificate is no longer needed. In other words, the CLS technique has the potential to address issues in both the classic signature method and the ID-based method. CLAS has the benefits of CLS and AS. In 2003, Boneh et al. [10] proposed the concept of CLAS, which can combine the signatures of n (n > 1) different messages signed by n other users into a single signature. The receiver only needs to check the aggregated signature instead of all the signatures, thereby reducing the computational cost of signature verification and the communication overhead of signature transmission to some extent.

The benefits of CLAS mentioned above have led to a lot of new research. Yum and Lee [12] proposed a CLAS scheme within the framework of the Random Oracle Model (ROM), but Hu et al. [13] discovered that Yum and Lee's [12] scheme is vulnerable to public key replacement attacks. Deng et al. [14] designed and proved a secure practical CLAS scheme, although Kumar and Sharma [15] found that Deng et al. [14]'s scheme could guarantee unforgeability. A new certificateless signature system was presented by Horng et al. [16] for the use of V2I in a VANET's communication. However, Ming and Shen [17] demonstrated that the scheme proposed by Horng et al. [16] was vulnerable to various attacks like replay attacks, modification attacks, impersonation attacks, and man-in-the-middle attacks and hence could not provide authentication and message integrity. Li et al. [18] addressed the limitation of Horng et al. [16]'s scheme and designed an improved CLAS scheme. However, the scheme has high computational and communicational costs since it uses bilinear pairing and point-to-point hash functions. Keitaro Hashimoto and Wakaha Ogata [19] came up with an open and small CLAS scheme in which the size of signatures stays the same, and any combination of signatures can be added together. However, the scheme is based on bilinear pairing, which necessitates higher computational and communicational costs. A highly effective AS scheme was developed by Malhi et al. [20] for privacy and authentication in VANETs. Cui et al. [21] developed an efficient CLAS scheme using ECC in vehicular sensor networks.

On the other hand, Kamil et al. [22] claimed that Cui et al.'s [21] scheme is unsafe against signature forgery attack. Du et al. [23] proposed an effective CLAS scheme without pairings for healthcare wireless sensor networks. However, the scheme is based on ECC, which results in a significant increase in both computational and communicational costs. To avoid the unpleasant certificate management problem of PKI and the key escrow problem of an ID-based framework, Gowri et al. [24] developed a CLAS-based authentication scheme for VANETS. However, Yang et al. [25] proved that Gowri et al.'s [24] scheme failed to achieve the desired security goals. Ye et al. [26] designed an improved certificateless authentication and AS scheme that may effectively counter coalition attacks. In the same year, Vallent et al. [27] developed a safe and efficient certificateless aggregation technique (ECLAS) for VANETs that might be used in a smart grid scenario. However, the [26,27] schemes are based on ECC to provide conditional privacy preservation, which leads to heavy computational costs and communication overhead.

A fully aggregated conditional privacy-preserving certificateless aggregate signature system (CPP-CLAS) was designed for VANETs by Yulei and Chen [28] in 2022. The proposed CPP-CLAS scheme uses ECC and general hash functions, which result in high computational costs and communication overhead. Another efficient pairing-free CLAS for secure VANET communication was introduced in the same year by Yibo et al. [29]. However, the [28,29] scheme was based on ECC, which has more computational costs and communication overhead.

In 2022, Cahyadi et al. [30] proposed a pairing-based CLAS authentication scheme to improve security, privacy, and efficiency in VANETs. However, it is found that the overall computation cost of the scheme is high due to massive pairing. To improve the security of VANET systems, we propose a CLAS-based authentication scheme based on HECC to reduce the computational cost and communication overhead due to the small parameter size.

## 3. Preliminaries

Koblitz introduced an algebraic curve called HEC [31]. It is considered an advanced version of elliptic curves. Points of an HEC cannot be obtained from the group [32]. In HECC, the additive Abelian group is calculated from the divisor ($\mathcal{D_P}$). HECC may be a more suitable option for low-resource environments due to its ability to provide the same level of security as ECC, bilinear pairing, and RSA while utilizing smaller parameters and key sizes [33]. Let $F_*$ be the algebraic closure of $F_\#$, which is the finite field. Suppose the genus of HECC over $F_\#$ is ($\bar{g}$ >1). Hyperelliptic curve *(H (Up))* is a generalized form of elliptic curves and the state *H (Up)* over a finite field is defined using Equation (1):

$$H:\ a^2 + h(\ell)a = t(\ell)modq \tag{1}$$

where ($\ell$, $a$) belongs to [$F_\# \times F_\#$], and polynomial f($\ell$) belongs to ($\ell$) with degree $\bar{g}$ and a monic one as f($\ell$) = $F_\#(\ell)$ with a degree 2 $\bar{g}$ +1. If there is no pair ($\ell$, $a$) belonging to [$F_\# \times F_\#$], such a curve is called non-singular.

### 3.1. Hyperelliptic Curve Discrete Logarithm Problem (HECDLP) Assumptions

Suppose £ ∈ {1,2,3..., n−1} and $\mathcal{D_P}$ is the divisor from HECC.

Let ω = £. $\mathcal{D_P}$, and calculating $\mathcal{D_P}$ from this equation is called the HECC discrete logarithm problem (HECDP).

### 3.2. Hyperelliptic Curve Computational Defi-Helman Problem (HECCDHP) Assumptions

Suppose, $\vec{\varnothing}$, Ω ∈ {1,2,3..., n−1} and $\mathcal{D_P}$ is the divisor from HEC.

Let ω = $\vec{\varnothing}$. $\mathcal{D_P}$, ¥ = Ω. $\vec{\varnothing}$. $\mathcal{D_P}$, and calculating $\vec{\varnothing}$ and Ω from △ and ¥ is called the HEC computational Defi–Helman problem (HCDH).

### 3.3. Network Model

In Figure 2, the network model of the proposed scheme is shown, which includes three entities: onboard unit (OBU), roadside unit (RSU), and department of transportation (DoT). The following are the explanatory steps:

- OBU: It is a 5G-enabled communication device fixed on a vehicle that can communicate with RSU and other OBUs. It is responsible for registering itself with the DoT by sending its identity in an encrypted form. The DoT first decrypts the received encrypted identity, generates a partial private key for this identity, and returns it to the OBU in an encrypted format using an insecure channel. Then, the OBU generates a private key and a public key, generates a signature on data, and sends it to the RSU via an open network.
- RSU: It is a 5G-enabled base station responsible for managing and conducting V-I communication. It is responsible for registering itself with the DoT by sending its identity in an encrypted form. The DoT generates a partial private key for this identity and returns it to the RSU in an encrypted format using an insecure channel. Then, the RSU can produce a complete private key and public key. When the RSU receives signed data from the OBU, it verifies the signature and either accepts the message or generates an error message depending on the results. RSU also works as a signature aggregator.
- DoT: The DoT is a reliable third party (TA) with significant processing power and storage capability. When the DoT is provided with the identities of OBU and RSU, it produces a partial private key pair and sends it back to the OBU and RSU in two packages in an encrypted form using an insecure channel. Then, both OBU and RSU create their remaining private and public keys for themselves.

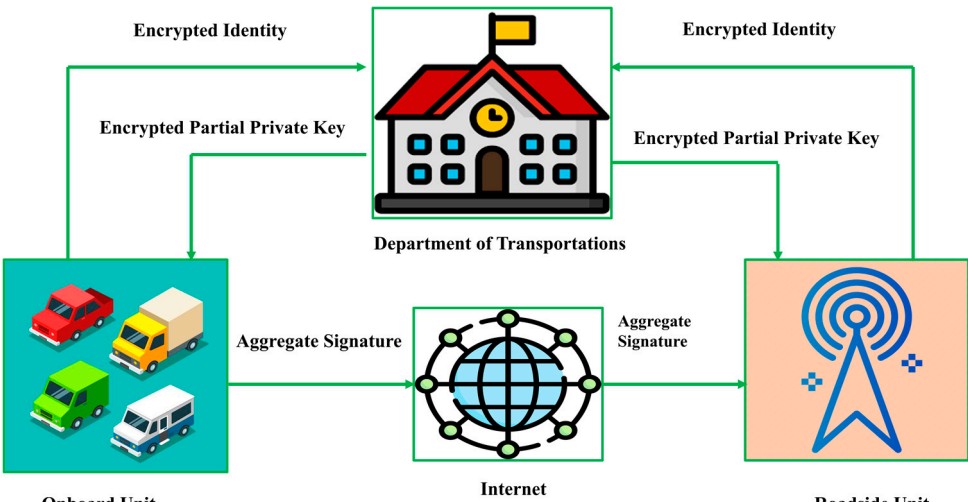

**Figure 2.** Network model of the proposed CLAS scheme.

### 3.4. Syntax of the Proposed CLAS Scheme

The syntax of our CLAS contains the following sub-steps:

*Setup:* The DoT can play the role of a TA, and it will be able to run this phase upon receiving the security parameter, in which it can first choose the hyperelliptic curve. Then, the DoT can set the private and public keys. Further, it generates public param and publishes it to the network.

*Partial Private Key Generation (PRPKG):* Any participating user who needs a partial private key can first encrypt their real identity using the common secret key between the DoT and that particular user. After doing this, the user sends an encrypted identity to the DoT using an insecure network. Then, the DoT decrypts the encrypted identity and recovers the real identity. Further, the DoT generates the partial private key for that particular identity, encrypts it, and delivers it to the user using an insecure network.

*Private Key Generation (PRKG):* In this section, the user generates his public and private key pairs.

*Signature Generation (SIGG):* This section will be run by the OBU to generate and send the signature tuple ($S_{OBU}, W_{OBU}$) to the RSU.

*Signature Verifications (SIGV):* This section will run by the RSU to verify the signature tuple ($S_{OBU}, W_{OBU}$).

### 4. Proposed Scheme's Construction

The proposed scheme consists of the steps listed below, and Table 1 outlines the fundamental symbols utilized in its construction. The construction of the proposed scheme is also illustrated in Figure 3.

**Table 1.** Symbols used for the construction of the proposed scheme.

| No | Notation | Descriptions |
|---|---|---|
| 1 | ($H_{yper}$) | *The hyperelliptic curve of genus 2* |
| 2 | $F^{ield}{}_p$ | *A finite field of the hyperelliptic curve with order* p |
| 3 | $Dot_p$ | *The private key of DoT* |
| 4 | $Dot_{pb}$ | *The public key of DoT* |
| 5 | D | *Divisor on hyperelliptic curve* |
| 6 | $H_{01}, H_{02}, H_{03}$ | *Hash Function with irreversibility* |

**Table 1.** *Cont.*

| No | Notation | Descriptions |
|---|---|---|
| 7 | $PB_{frm}$ | *Public parameter (param)* |
| 8 | $U_{sr}$ | *Represents the participating user* |
| 9 | $G_{usr}$ | *DoT, the random value selected by user* |
| 10 | $K_{usr}$ | *A secret shared key between user and DoT* |
| 11 | $EID_{usr}$ | *Encrypted identity of user* |
| 12 | $(F_{usr}, L_{usr})$ | *Public key pair of users* |
| 13 | $(G_{usr}, P_{usr})$ | *Private key pair of users* |
| 14 | $S_{OBU}$ | *Represent signature generated by OBU* |
| 15 | $P_{usr}$ | *Partial Private key of users* |
| 16 | $E_{K_{usr}}$ | *Represents an encryption procedure* |
| 17 | $D_{K_{usr}}$ | *Represents the decryption procedure* |
| 18 | $ID_{usr}$ | *Identity of user* |

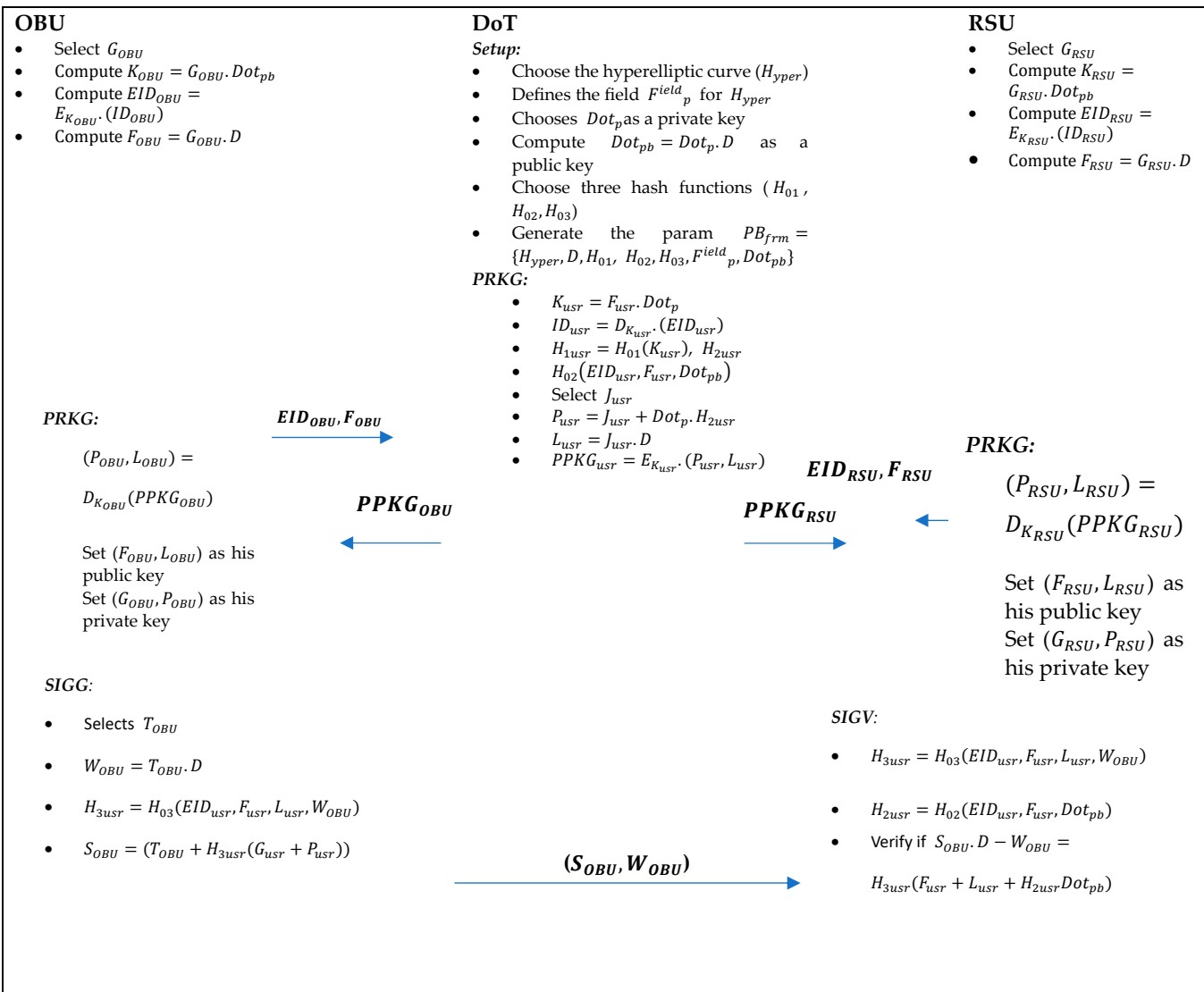

**Figure 3.** Construction of the proposed CLAS scheme.

*Setup:* The DoT can play the role of a TA, and it will be able to run this phase upon receiving the security parameter $k$, in which it can first choose the hyperelliptic curve ($H_{yper}$) that defines field $F^{ield}_p$, a devisor of 80 bis, and then it chooses $Dot_p$ as a private key from $F^{ield}_p$ and computes $Dot_{pb} = Dot_p.D$ as a public key. Further, it can choose three hash functions, $H_{01}$, $H_{02}$, *and* $H_{03}$ and the public param $PB_{frm} = \{H_{yper}, D, H_{01}, H_{02}, H_{03}, F^{ield}_p, Dot_{pb}\}$.

*Partial Private Key Generation (PRPKG):* Any participating user $U_{sr}$ who needs a partial private key can first select $G_{usr}$ from $F^{ield}_p$, compute $K_{usr} = G_{usr}.Dot_{pb}$, calculate $EID_{usr} = E_{K_{usr}}.(ID_{usr})$, compute $F_{usr} = G_{usr}.D$, and send $(EID_{usr}, F_{usr})$ to DoT through the internet. Upon receiving $(EID_{usr}, F_{usr})$, DoT can compute $K_{usr} = F_{usr}.Dot_p$, recover user identity as $ID_{usr} = D_{K_{usr}}.(EID_{usr})$, compute $H_{1usr} = H_{01}(K_{usr})$, $H_{2usr} = H_{02}\left(EID_{usr}, F_{usr}, Dot_{pb}\right)$, select $J_{usr}$ from $F^{ield}_p$, compute $P_{usr} = J_{usr} + Dot_p.H_{2usr}$, calculate $L_{usr} = J_{usr}.D$, encrypt $PPKG_{usr} = E_{K_{usr}}.(P_{usr}, L_{usr})$, and send $PPKG_{usr}$ to users through an open channel.

*Private Key Generation (PRKG):* Upon receiving $PPKG_{usr}$, any participating user $U_{sr}$ can recover $(P_{usr}, L_{usr})$ as $(P_{usr}, L_{usr}) = D_{K_{usr}}(PPKG_{usr})$ and set $(F_{usr}, L_{usr})$ as his public key and $(G_{usr}, P_{usr})$ as his private key.

*Signature Generation (SIGG):* This section will be run by the OBU using the following steps:

- It selects $T_{OBU}$ from $F^{ield}_p$ and computes $W_{OBU} = T_{OBU}.D$;

- It computes $H_{3usr} = H_{03}(EID_{usr}, F_{usr}, L_{usr}, W_{OBU})$;

- It computes $S_{OBU} = (T_{OBU} + H_{3usr}(G_{usr} + P_{usr}))$ and sends $(S_{OBU}, W_{OBU})$ to the RSU.

*Signature Verifications (SIGV):* This section will be run by the RSU using the following steps:

- Computes $H_{3usr} = H_{03}(EID_{usr}, F_{usr}, L_{usr}, W_{OBU})$ and $H_{2usr} = H_{02}\left(EID_{usr}, F_{usr}, Dot_{pb}\right)$;

- Verifies if $S_{OBU}.D - W_{OBU} = H_{3usr}\left(F_{usr} + L_{usr} + H_{2usr}Dot_{pb}\right)$, if it is satisfied.

Aggregate Signature Generation and Verifications: This part is the same as performed in [29].

*Adding New Device:* If a new device wants to add itself to the network in our proposed scheme, it will first perform the following procedures: it can first select $G_{new}$ from $F^{ield}_p$, compute $K_{new} = G_{new}.Dot_{pb}$, calculate $EID_{new} = E_{K_{new}}.(ID_{new})$, compute $F_{new} = G_{new}.D$, and send $(EID_{new}, F_{new})$ to DoT through the internet. Upon receiving $(EID_{new}, F_{new})$, DoT can compute $K_{new} = F_{new}.Dot_p$, recover new device identity as $ID_{new} = D_{K_{new}}.(EID_{new})$, compute $H_{1new} = H_{01}(K_{new})$, $H_{2new} = H_{02}\left(EID_{new}, F_{new}, Dot_{pb}\right)$, select $J_{new}$ from $F^{ield}_p$, compute $P_{new} = J_{new} + Dot_p.H_{2new}$, calculate $L_{new} = J_{new}.D$, encrypt $PPKG_{new} = E_{K_{new}}(P_{new}, L_{new})$, and send $PPKG_{new}$ to new devices through an open channel. A new device then recovers $(P_{new}, L_{new})$ as $(P_{new}, L_{new}) = D_{K_{new}}(PPKG_{new})$, sets $(F_{new}, L_{new})$ as his public key, sets $(G_{new}, P_{new})$ as his private key, and he can start communications with other devices if he wants.

*Correctness*

RSU will verify the signature if the following steps are satisfied.

$$S_{OBU}.D - W_{OBU} = H_{3usr}\left(F_{usr} + L_{usr} + H_{2usr}Dot_{pb}\right)$$

$S_{OBU}.D - W_{OBU} = (T_{OBU} + H_{3usr}(G_{usr} + P_{usr})).D - W_{OBU} = (T_{OBU}.D + H_{3usr}(G_{usr} + P_{usr}).D) - W_{OBU} = (W_{OBU} + H_{3usr}(G_{usr} + P_{usr}).D) - W_{OBU} = (H_{3usr}(G_{usr} + P_{usr}).D) = (H_{3usr}(G_{usr}.D + P_{usr}.D)) = (H_{3usr}(G_{usr}.D + (J_{usr} + Dot_p.H_{2usr}).D)) = (H_{3usr}(G_{usr}.D + (J_{usr}.D + Dot_p.H_{2usr}.D))) = (H_{3usr}(G_{usr}.D + (J_{usr}.D + Dot_p.D.H_{2usr}))) = H_{3usr}(F_{usr} + L_{usr} + H_{2usr}Dot_{pb})$, hence proved.

## 5. Security Analysis

In this phase, with the help of Theorems 1 and 2, we are going to prove that the proposed aggregate signature scheme is unforgeable against Type 1 ($FGR_1$) and Type 2 ($FGR_2$) forgers.

**Theorem 1.** *Utilizing a non-negligible probability $n_{Onpb}$, if Type 1 ($FGR_1$) forger wants to forge the proposed certificateless aggregate signature within a polynomial time successfully, then there will be a challenger (FCR) who can serve his services as a facilitator for solving the hyperelliptic curve discrete logarithm problem with the probability $\left(1 - \frac{1}{X}\right)\left(\frac{n_{Onpb}}{XQ_{H_{usri}}}\right)\left(1 - \frac{1}{Q_{PRPKG} + Q_{SIGG} + Q_{SIGV} + 1}\right)$, where $Q_{PRPKG}, Q_{SIGG}, Q_{SIGV}, X, n_{Onpb}$ and $Q_{H_{usri}}$ represent Partial Private Key Generation Query, Signature Generation Query, Signature Verifications Query, Natural Logarithm, Non-negligible Probability, and Hash Queries, respectively.*

**Proof.** A facilitator called FCR will solve the hyperelliptic curve discrete logarithm problem if he receives $D, HDLP = x.D$ and his task will be to extract $x$ from $HDLP$. □

**Setup:** FCR calls the Setup algorithm in which it sends the param $PB_{frm}$ to $FGR_1$ and keeps private $Dot_p$.

**Query Phase:** $FGR_1$ selects the identity $ID_{usri}$ and FCR has no access to $ID_{usri}$. Then, *FCR* selects the identity $ID_{usri}{}^*$ and performs the following queries with $FGR_1$:

**$H_{01}$ Query:** FCR initializes an empty list ($L_{H_{1usr}}$). When $FGR_1$ generates this query, *FCR* checks the tuple ($K_{usr}, H_{1usr}$) in $L_{H_{1usr}}$. If the value $H_{1usr}$ exists, then it sends $H_{1usr}$ to $FGR_1$; otherwise, it randomly picks $H_{1usr}$ from $F^{ield}{}_p$, sends $H_{1usr}$ to $FGR_1$ and updates $L_{H_{1usr}}$ with the value $H_{1usr}$.

**$H_{02}$ Query:** FCR initializes an empty list ($L_{H_{2usr}}$). When $FGR_1$ generates this query, *FCR* checks the tuple $\left(EID_{usr}, F_{usr}, Dot_{pb}, H_{2usr}\right)$ in $L_{H_{2usr}}$. If the value $H_{2usr}$ exists, then it sends $H_{2usr}$ to $FGR_1$; otherwise, it randomly picks $H_{2usr}$ from $F^{ield}{}_p$, sends $H_{2usr}$ to $FGR_1$ and updates $L_{H_{2usr}}$ with the value $H_{2usr}$.

**$H_{03}$ Query:** FCR initializes an empty list ($L_{H_{3usr}}$). When $FGR_1$ generates this query, *FCR* checks the tuple ($EID_{usr}, F_{usr}, L_{usr}, W_{OBU}, H_{3usr}$) in $L_{H_{3usr}}$. If the value $H_{3usr}$ exists, then it sends $H_{3usr}$ to $FGR_1$; otherwise, it randomly picks $H_{3usr}$ from $F^{ield}{}_p$, sends $H_{3usr}$ to $FGR_1$ and updates $L_{H_{3usr}}$ with the value $H_{3usr}$.

**Secret Value Generation (SVG) Query:** FCR initializes an empty list ($L_{SVG}$). When $FGR_1$ generates this query, *FCR* checks the value ($G_{usr}$) in $L_{SVG}$. If the value $G_{usr}$ exists, then it sends $G_{usr}$ to $FGR_1$; otherwise, it randomly picks $G_{usr}$ from $F^{ield}{}_p$, sends $G_{usr}$ to $FGR_1$ and updates $L_{SVG}$ with the value $G_{usr}$.

**PRPKG Query:** FCR initializes an empty list ($L_{PRPKG}$). When $FGR_1$ generates this query, *FCR* checks the value ($P_{usr}$) in $L_{PRPKG}$. If the value $P_{usr}$ exists, then it sends $P_{usr}$ to $FGR_1$; otherwise, it randomly picks $P_{usr}$ from $F^{ield}{}_p$, sends $P_{usr}$ to $FGR_1$ and updates $L_{PRPKG}$ with the value $P_{usr}$.

**Public Key Generation (PBKG) Query:** FCR initializes an empty list ($L_{PBKG}$). When $FGR_1$ generates this query, *FCR* checks the value ($F_{usr}, L_{usr}$) in $L_{PBKG}$. If the tuple ($F_{usr}, L_{usr}$) exists, then it sends ($F_{usr}, L_{usr}$) to $FGR_1$; otherwise, it performs the following steps:

If $ID_{usri}! = ID_{usri}{}^*$, it chooses $G_{usr}, P_{usr}, H_{2usr}$ from $F^{ield}{}_p$, computes $F_{usr} = G_{usr}.D$, and calculates $L_{usr} = P_{usr}.D - H_{2usr}.Dot_{pb}$. Then, it sends ($F_{usr}, L_{usr}$) to $FGR_1$ and updates $L_{PBKG}$ with the value ($F_{usr}, L_{usr}$);

If $ID_{usri} = ID_{usri}{}^*$, it chooses $G_{usr}, J_{usr}$ from $F^{ield}{}_p$, computes $F_{usr} = G_{usr}.D$, and calculates $L_{usr} = J_{usr}.D$. Then, it sends ($F_{usr}, L_{usr}$) to $FGR_1$ and updates $L_{PBKG}$ with the value ($F_{usr}, L_{usr}$).

**Public Key Replaced (PKR) Query:** FGR$_1$ chooses the new public key tuple (F$_{usr}$*, L$_{usr}$*) and sends it to the FCR; then, FCR includes (F$_{usr}$*, L$_{usr}$*) into L$_{PBKG}$ as a replacement of the public key.

**SIGG Query:** FCR initializes an empty list (L$_{SIGG}$). When $FGR_1$ generates this query, $FCR$ checks the value $(ID_{usri}, m, G_{usr}, P_{usr})$ in $L_{SIGG}$. If the value $P_{usr}$ exists, then it selects from $F^{ield}{}_p$ , computes $W_{OBU} = T_{OBU}.D$, computes $H_{3usr} = H_{03}(EID_{usr}, F_{usr}, L_{usr}, W_{OBU})$, computes $S_{OBU} = (T_{OBU} + H_{3usr}(G_{usr} + P_{usr}))$ and sends $(S_{OBU}, W_{OBU})$ to $FGR_1$. Otherwise, it selects $S_{OBU}$ from $F^{ield}{}_p$ and sends it to the $FGR_1$.

**Forgery:** When the above queries are completed successfully, FGR$_1$ can return a forged certificateless signature tuple $(S_{OBU}{}^*, W_{OBU}{}^*)$. By using the concept of the forking lemma, $FGR_1$ can return another forged certificateless signature tuple $(S_{OBU}{}^{*1}, W_{OBU}{}^{*1})$. So these two tuples will be only true if $FCR$ gets the valid value of $x$.

To satisfy this theorem, the results generated via $FCR$ must meet the following conditions:
$FCR_{C1}$ : $FCR$ does not stop the querying process and its probability $(1 - \frac{1}{X})$.
$FCR_{C1}$ : $FCR$ does not stop the forging process for signature and its probability $\left( \frac{n_{Onpb}}{XQ_{H_{usri}}} \right)$.
$FCR_{C1}$ : $(S_{OBU}{}^*, W_{OBU}{}^*)$ is a valid tuple, and its probability $\left( 1 - \frac{1}{Q_{PRPKG} + Q_{SIGG} + Q_{SIGV} + 1} \right)$.
So, the probability of $(FCR_{C1}.FCR_{C2}.FCR_{C2})$ as $\left( (1 - \frac{1}{X}) \left( \frac{n_{Onpb}}{XQ_{H_{usri}}} \right) \left( 1 - \frac{1}{Q_{PRPKG} + Q_{SIGG} + Q_{SIGV} + 1} \right) \right)$.

**Theorem 2.** *Utilizing a non-negligible probability $n_{Onpb}$, if Type 2 ($FGR_2$) forger wants to forge our certificateless aggregate signature within a polynomial time successfully, then there will be a challenger ($FCR$) who can serve his services as a facilitator for solving the hyperelliptic curve discrete logarithm problem with the probability* $\left( 1 - \frac{1}{X} \right) \left( \frac{n_{Onpb}}{XQ_{H_{usri}}} \right) \left( 1 - \frac{1}{Q_{SIGG} + Q_{SIGV} + 1} \right)$, *where $Q_{SIGG}, Q_{SIGV}, X, n_{Onpb}$ and $Q_{H_{usri}}$ represent Signature Generation Query, Signature Verifications Query, Natural Logarithm, Non-negligible Probability, and Hash Queries, respectively.*

**Proof.** A facilitator called FCR will solve the hyperelliptic curve discrete logarithm problem if he receives $D, HDLP = x.D$ and his task will be to extract $x$ from $HDLP$. □

**Setup:** FCR calls the Setup algorithm in which he sends the param PB$_{frm}$ and Dot$_p$ to FGR$_2$

**Query Phase:** FGR$_2$ selects the identity ID$_{usri}$ and FCR has no access to ID$_{usri}$. Then, *FCR* selects the identity $ID_{usri}{}^*$ and performs the following queries with $FGR_2$:

**H$_{01}$ Query:** FCR sets a list (L$_{H_{1usr}}$), which is empty. When FGR$_2$ generates this query, FCR checks the tuple in L$_{H_{1usr}}$. If the value H$_{1usr}$ exists, then it sends H$_{1usr}$ to FGR$_1$; otherwise, it randomly picks H$_{1usr}$ from F$^{ield}{}_p$, sends H$_{1usr}$ to FGR$_2$ and updates $L_{H_{1usr}}$ with the value H$_{1usr}$.

**H$_{02}$ Query:** FCR sets an empty list (L$_{H_{2usr}}$). When $FGR_2$ generates this query, $FCR$ checks the tuple $\left( EID_{usr}, F_{usr}, Dot_{pb}, H_{2usr} \right)$ in $L_{H_{2usr}}$. If the value $H_{2usr}$ exists, then it sends $H_{2usr}$ to $FGR_2$; otherwise, it randomly picks $H_{2usr}$ from $F^{ield}{}_p$, sends $H_{2usr}$ to $FGR_2$ and updates $L_{H_{2usr}}$ with the value $H_{2usr}$.

**H$_{03}$ Query:** FCR sets an empty list (L$_{H_{3usr}}$). When $FGR_2$ generates this query, $FCR$ checks the tuple $(EID_{usr}, F_{usr}, L_{usr}, W_{OBU}, H_{3usr})$ in $L_{H_{3usr}}$. If the value $H_{3usr}$ exists, then it sends $H_{3usr}$ to $FGR_2$; otherwise, it randomly picks $H_{3usr}$ from $F^{ield}{}_p$, sends $H_{3usr}$ to $FGR_2$ and updates $L_{H_{3usr}}$ with the value $H_{3usr}$.

**Secret Value Generation (SVG) Query:** FCR sets an empty list (L$_{SVG}$). When $FGR_2$ generates this query, $FCR$ checks the value $(G_{usr})$ in $L_{SVG}$. If the value exists, then it

sends $G_{usr}$ to $FGR_2$; otherwise, it randomly picks $G_{usr}$ from $F^{ield}{}_p$, sends $G_{usr}$ to $FGR_2$ and updates $L_{SVG}$ with the value $G_{usr}$.

**Public key Generation (PBKG)Query:** FCR sets an empty list ($L_{PBKG}$). When $FGR_2$ generates this query, $FCR$ checks the value $(F_{usr}, L_{usr})$ in $L_{PBKG}$. If the tuple $(F_{usr}, L_{usr})$ exists, then it sends $(F_{usr}, L_{usr})$ to $FGR_2$; otherwise, it performs the following steps:

If $ID_{usri}! = ID_{usri}{}^*$, it chooses $G_{usr}, P_{usr}, H_{2usr}$ from $F^{ield}{}_p$, computes $F_{usr} = G_{usr}.D$, and calculates $L_{usr} = P_{usr}.D - H_{2usr}.Dot_{pb}$. Then, it sends $(F_{usr}, L_{usr})$ to $FGR_2$ and updates $L_{PBKG}$ with the value $(F_{usr}, L_{usr})$;

If $ID_{usri} = ID_{usri}{}^*$, it chooses $G_{usr}, J_{usr}$ from $F^{ield}{}_p$, computes $F_{usr} = G_{usr}.D$, and calculates $L_{usr} = J_{usr}.D$. Then, it sends $(F_{usr}, L_{usr})$ to $FGR_2$ and updates $L_{PBKG}$ with the value $(F_{usr}, L_{usr})$.

*SIGG Query:* FCR sets an empty list ($L_{SIGG}$). When $FGR_2$ generates this query, $FCR$ checks the value $(ID_{usri}, m, G_{usr}, P_{usr})$ in $L_{SIGG}$. If the value $P_{usr}$ exists, then it selects $T_{OBU}$ from $F^{ield}{}_p$ and computes $W_{OBU} = T_{OBU}.D$, computes $H_{3usr} = H_{03}(EID_{usr}, F_{usr}, L_{usr}, W_{OBU})$, computes $S_{OBU} = (T_{OBU} + H_{3usr}(G_{usr} + P_{usr}))$ and sends $(S_{OBU}, W_{OBU})$ to $FGR_1$. Otherwise, it selects $S_{OBU}$ from $F^{ield}{}_p$ and sends it to $FGR_2$.

*Forgery:* When the above queries are completed successfully, $FGR_1$ can return a forged certificateless signature tuple $(S_{OBU}{}^*, W_{OBU}{}^*)$. By using the concept of the forking lemma, $FGR_1$ can return another forged certificateless signature tuple $(S_{OBU}{}^{*1}, W_{OBU}{}^{*1})$. So these two tuples will be only true if $FCR$ gets the valid value of $x$.

To satisfy this theorem, the results generated via $FCR$ must meet the following conditions:

$FCR_{C1}$ : does not stop the querying process and its probability $(1 - \frac{1}{X})$.

$FCR_{C1}$ : $FCR$ does not stop the forging process for signature and its probability $\left(\frac{n_{Onpb}}{XQ_{H_{usri}}}\right)$.

$FCR_{C1}$ : $(S_{OBU}{}^*, W_{OBU}{}^*)$ is a valid tuple and its probability $\left(1 - \frac{1}{Q_{SIGG} + Q_{SIGV} + 1}\right)$.

So, the probability of $(FCR_{C1}.FCR_{C2}.FCR_{C2})$ as $\left((1 - \frac{1}{X})\left(\frac{n_{Onpb}}{XQ_{H_{usri}}}\right)\left(1 - \frac{1}{Q_{SIGG} + Q_{SIGV} + 1}\right)\right)$.

## 6. Performance Comparison

This section compares the proposed scheme with relevant schemes in terms of security requirements, computational cost, and communication overhead.

### 6.1. Security Requirements Comparisons

In this section, we have compared our proposed scheme with the existing scheme based on security requirements that are unforgeability against FGR$_1$(UF1), unforgeability against FGR$_2$(UF2), sender anonymity (SA) at the time of request generation for the partial private key, receiver anonymity (RA) at the time of request generation for the partial private key, and removal of the concept of the secure channel during distribution of the partial private key (PPK). Therefore, we used YES if the scheme satisfies the security requirement and NO if otherwise. Based on Table 2, we can conclude that the proposed scheme satisfies all of the above security claims. In contrast, existing schemes such as Cahyadi et al. [30], Keitaro and Ogata [19], Yulei and Chen [28], and Yibo et al. [29] do not meet the SA, RA, and PPK requirements.

**Table 2.** Security attributes comparison.

| Scheme | UF1 | UF2 | SA | RA | PPK |
|---|---|---|---|---|---|
| Eko Cahyadi et al. [30] | YES | YES | NO | NO | NO |
| Yulei and Chen [28] | YES | YES | NO | NO | NO |
| Yibo et al. [29] | YES | YES | NO | NO | NO |
| Keitaro and Ogata [19] | YES | YES | NO | NO | NO |
| Proposed | YES | YES | YES | YES | YES |

### 6.2. Computational Cost

The computational cost reveals how much processing is required on both the sender and receiver ends of the communication. Major procedures like modular exponential, bilinear pairing operations, elliptic curve point multiplication, and hyperelliptic curve divisor scalar multiplication are used to calculate the computational cost. In this section, we have compared our proposed CLAS scheme in terms of computational cost with the relevant schemes of Cahyadi et al. [30], Keitaro and Ogata [19], Yulei and Chen [28], and Yibo et al. [29], as depicted in Table 3. For this purpose, we then show that $\mathcal{B}^\mathbb{P}\mathcal{M}$, $\mathcal{B}^\mathbb{P}$, $\mathcal{EC}$, $\mathcal{H}\mathcal{EC}$ denote bilinear point multiplication, bilinear pairing, elliptic curve multiplication, and hyperelliptic curve multiplication, respectively.

**Table 3.** Comparison of computational cost for relevant CLAS schemes.

| Scheme | Signing Cost | Verification Cost | Total |
|---|---|---|---|
| Eko Cahyadi et al. [30] | $5\,\mathcal{B}^\mathbb{P}\mathcal{M}$ | $3\,\mathcal{B}^\mathbb{P}$ | $8\,\mathcal{B}^\mathbb{P}$ |
| Yulei and Chen [28] | $2\,\mathcal{EC}$ | $3\,\mathcal{EC}$ | $5\,\mathcal{EC}$ |
| Yibo et al. [29] | $2\,\mathcal{EC}$ | $2\,\mathcal{EC}$ | $4\,\mathcal{EC}$ |
| Keitaro and Ogata [19] | $3\,\mathcal{B}^\mathbb{P}\mathcal{M}$ | $4\,\mathcal{B}^\mathbb{P}$ | $7\,\mathcal{B}^\mathbb{P}$ |
| Proposed | $2\,\mathcal{H}\mathcal{EC}$ | $2\,\mathcal{H}\mathcal{EC}$ | $4\,\mathcal{H}\mathcal{EC}$ |

The execution time for several time-consuming cryptographic operations is summarized in Table 4 [33]. The experiment is conducted through Intel Core i5-6300 CPU, 2.40 GHz processor, 8 GB of RAM, Windows 10 Ultimate edition, and Multiprecision Integer and Rational Arithmetic C Library (MIRACL). Then, the following formulas are used to compute the computational cost of the proposed CLAS scheme in milliseconds (ms) with the relevant schemes of Cahyadi et al. [30], Keitaro and Ogata [19], Yulei and Chen [28], and Yibo et al. [29], as presented in Table 5:

- For bilinear pairing-based scheme, we used the following formulas for computational cost:

*Signing Cost* = Number of Bilinear Pairing Operations ∗ Time required

for Single Bilinear Paring Operation

*Verification Cost* = Number of Bilinear Pairing Operations ∗ Time required for Single

Bilinear Paring Operation

**Table 4.** Execution time of different cryptographic operations.

| Operation | Bilinear Point Multiplication | Bilinear Pair | Elliptic Curve | Hyper Elliptic Curve |
|---|---|---|---|---|
| **Time in ms** | 4.31ms | 14.90ms | 0.97ms | 0.48ms |

**Table 5.** Comparison of computational cost based on major operations in ms.

| Scheme | Signing Cost | Verifying Cost | Total |
|---|---|---|---|
| Eko Cahyadi et al. [30] | 21.55 | 44.7 | 66.25 |
| Yulei and Chen [28] | 1.94 | 2.91 | 4.85 |
| Yibo et al. [29] | 1.94 | 1.94 | 3.88 |
| Keitaro and Ogata [19] | 12.93 | 59.6 | 72.53 |
| Proposed | 0.96 | 0.96 | 1.92 |

- For elliptic curve-based scheme, we used the following formulas for computational cost:

*Signing Cost* = Number of Elliptic Curve Operations ∗ Time

required for Single Elliptic Curve Operation

*Verification Cost* = Number of Elliptic Curve Operations ∗ Time required for

Single Elliptic Curve Operation

- For hyperelliptic curve-based scheme, we used the following formulas for computational cost:

*Signing Cost* = Number of Hyper Elliptic Curve Operations ∗ Time

required for Single Hyper Elliptic Curve Operation

*Verification Cost* = Number of Hyper Elliptic Curve Operations ∗ Time required for

Single Hyper Elliptic Curve Operation

Therefore, Table 5 shows that the new method uses less computing power by using hyperelliptic curve cryptography with a key size of only 80 bits and provides the same level of security as RSA and elliptic curve cryptography. Figure 4 depicts that our scheme outperforms [19,28–30].

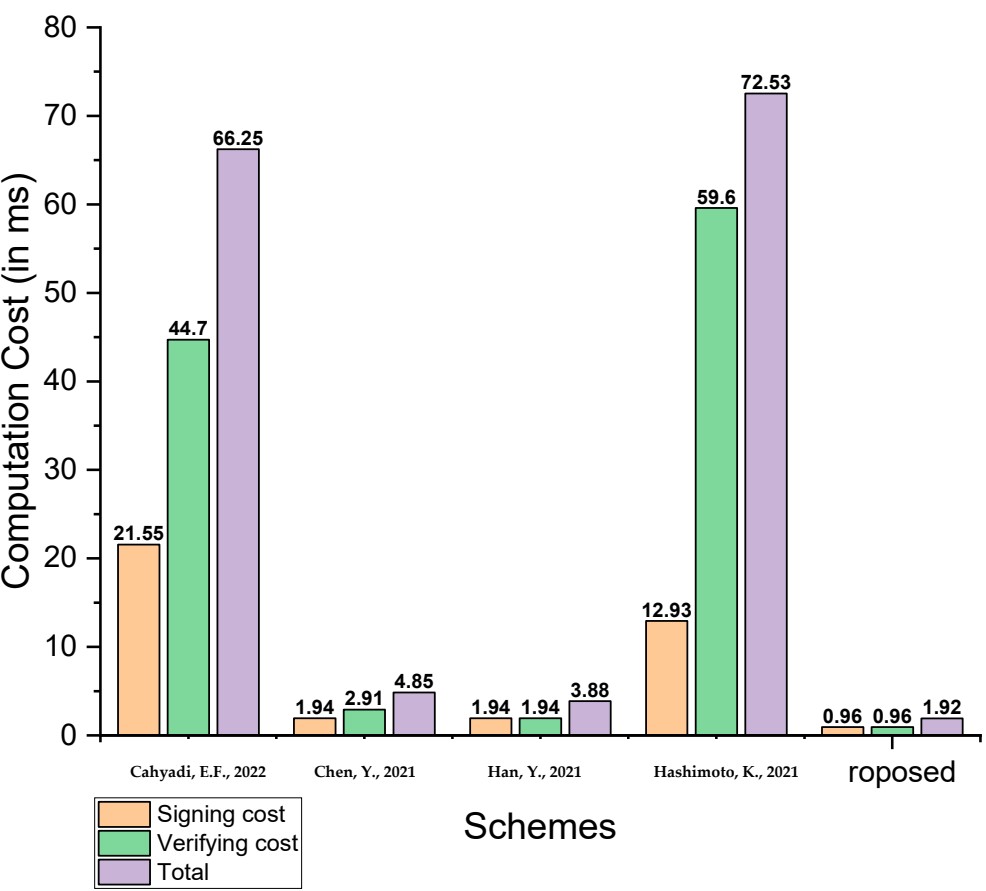

**Figure 4.** Computation cost (in ms) [19,28–30].

*6.3. Communication Cost*

The term "communication cost" refers to the extra bits added to the message. In this section, we compare our proposed scheme with relevant schemes offered by Eko

Cahyadi et al. [30], Keitaro and Ogata [19], Yulei and Chen [28], and Yibo et al. [29], as depicted in Table 3, in terms of communicational overhead. For this purpose, we picked the values from [33] that are plaintext ($|m|$) and has a size of 1000 bits, while the size of the bilinear pairing ($|G|$) is 1024 bits, the size of the elliptic curve ($|q|$), is 160 bits, and the size of the hyperelliptic curve ($|n|$) is 80 bits, respectively. As demonstrated in Table 6 and Figure 5, our proposed technique provides more significant improvements over the schemes used in [19,28–30]. Note that we have used the following formulas to compute the communication cost based on transmitted bits:

- Communicational cost formula for bilinear pairing-based schemes: $|\text{Message}| + |\text{Total number of Transmitted parameters}| * 1024$.
- Communicational cost formula for elliptic curve schemes: $|\text{Message}| + |\text{Total number of Transmitted parameters}| * 160$.
- Communicational cost formula for hyperelliptic curve-based schemes: $|\text{Message}| + |\text{Total number of Transmitted parameters}| * 80$.

**Table 6.** Communication cost of different CLAS schemes.

| Scheme | Communication Cost | Communication Cost with Bits |
|---|---|---|
| Eko Cahyadi et al. [30] | $|m| + 5|G|$ | $1000 + 5 \times 1024 = 6120$ |
| Yulei and Chen [28] | $|m| + 5|q|$ | $1000 + 5 \times 160 = 1800$ |
| Yibo et al. [29] | $m| + 5|q|$ | $1000 + 5 \times 160 = 1800$ |
| Keitaro and Ogata [19] | $|m| + 2|G|$ | $1000 + 2 \times 1024 = 3048$ |
| Proposed | $|m| + 2|n|$ | $1000 + 2 \times 80 = 1160$ |

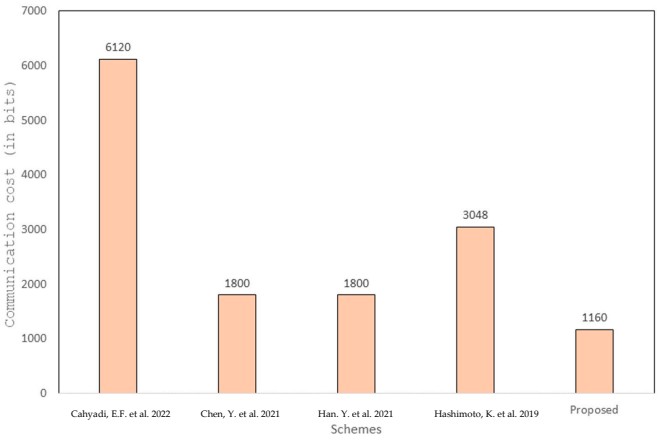

**Figure 5.** Communication cost (in bits) [19,28–30].

## 7. Conclusions

This paper presented a certificateless aggregate signature (CLAS) scheme based on hyperelliptic curve cryptography (HECC) that guarantees sender and receiver anonymity. Typically, anonymity is ensured by requiring participants to transmit their identities in an encrypted format when requesting a partial private key from the DoT/TA. To address the issue of a secure channel for certificateless cryptography, the DoT/TA can transmit the partial private key to participating users over an unencrypted, open channel. Through the provable security analysis, we demonstrated that our scheme is resistant to Type 1 ($FGR_1$) and Type 2 ($FGR_2$) forgery. During the random oracle model (ROM) provable security analysis, the proposed scheme relied on the hyperelliptic curve discrete logarithm problem. The proposed scheme outperforms the existing scheme in terms of computation and communication cost based on a comparison with the existing schemes that was performed to evaluate our scheme's efficiency. In future work, we plan to design a multi-sender and multi-receiver certificateless aggregate signature for VANETs. Further, the new scheme

will be based on the genus 3 hyperelliptic curve, which can be more efficient in terms of communication and computational costs.

**Author Contributions:** Conceptualization, A.I., M.A.K., A.V.S., I.U. and G.U.-R.; Methodology, M.A.K., A.I and A.V.S.; Software, I.U., M.A.K. and M.Z.; Validation, M.A.K., M.Z. and I.U.; Formal analysis, I.U., F.N. and M.A.K.; Investigation, A.I., A.V.S., I.U. and G.U.-R.; Resources, M.A.K., M.Z. and A.V.S.; Data curation, M.Z., A.V.S., F.N., G.U.-R. and A.I; Writing—original draft preparation, M.A.K., I.U., A.I., F.N., A.V.S. and M.Z.; Writing—review and editing, M.A.K., A.I., M.Z., F.N. and A.V.S.; Visualization, M.A.K., M.Z., G.U.-R., F.N. and I.U.; Supervision, M.Z. All authors have read and agreed to the published version of the manuscript.

**Funding:** This research received no external funding.

**Data Availability Statement:** Not applicable.

**Conflicts of Interest:** The authors declare no conflict of interest.

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
