# Peer review of "An Efficient and Secure Certificateless Aggregate Signature Scheme for Vehicular Ad hoc Networks"

_futureinternet, doi:10.3390/fi15080266_

Round 1
Reviewer 1 Report
Vehicular ad-hoc networks (VANETs) have become an essential part of the intelligent transportation system because they provide secure communication among the vehicles, enhance vehicle safety, and improve the driving experience. However, due to the openness and vulnerability of wireless networks, the participating vehicles in a VANET system are prone to a variety of cyber attacks. This paper attempted to propose a signature scheme to secure the privacy of vehicles and assure the authenticity, integrity, and nonrepudiation of messages. To enable secure communication between the vehicle and the roadside unit (RSU), this paper proposed a certificateless aggregate signature (CLAS) scheme based on the hyperelliptic curve cryptography (HECC). This scheme enables participating vehicles to share their identities with trusted authorities via an open wireless channel without revealing their identities to unauthorized participants. Numerical results showed advantages of the proposed scheme. The paper is well written but the following issues are recommended to be discussed to improve the paper's academic quality.
1) In Sect. 6 of performance comparison, there is no evaluation and performance comparison of the proposed scheme against conventional schemes, in terms of guaranteeing security, one of the main KPIs of the proposed system.
2) Please discuss about the scalability of the proposed mechanism against increasing number of vehicles. It is not clear how large the size of the system that the authors considered.
Author Response
Vehicular ad-hoc networks (VANETs) have become an essential part of the intelligent transportation system because they provide secure communication among the vehicles, enhance vehicle safety, and improve the driving experience. However, due to the openness and vulnerability of wireless networks, the participating vehicles in a VANET system are prone to a variety of cyber-attacks. This paper attempted to propose a signature scheme to secure the privacy of vehicles and assure the authenticity, integrity, and nonrepudiation of messages. To enable secure communication between the vehicle and the roadside unit (RSU), this paper proposed a certificateless aggregate signature (CLAS) scheme based on the hyperelliptic curve cryptography (HECC). This scheme enables participating vehicles to share their identities with trusted authorities via an open wireless channel without revealing their identities to unauthorized participants. Numerical results showed advantages of the proposed scheme. The paper is well written but the following issues are recommended to be discussed to improve the paper's academic quality.
Response: We would like to express our sincere appreciation for the reviewer's valuable time and diligent effort in evaluating our research paper. We appreciate the opportunity to resolve your concerns and enhance our work in light of the helpful suggestions provided by our esteemed reviewer. Following careful consideration of the feedback received, the following changes have been made to the paper:
Comment 1: In Sect. 6 of performance comparison, there is no evaluation and performance comparison of the proposed scheme against conventional schemes, in terms of guaranteeing security, one of the main KPIs of the proposed system.
Response 1: Indeed, we neglected a crucial KPI, so we are grateful to the esteemed reviewer for bringing this to our attention, which prompted us to compare the security properties of the proposed scheme with the relevant existing schemes. We added a section 6.1 titled "Security Requirements Comparisons." We hope the reservations made by reviewer have been honored.
Comment 2: Please discuss about the scalability of the proposed mechanism against increasing number of vehicles. It is not clear how large the size of the system that the authors considered.
Response 2: We are grateful to the reviewer for this insightful comment, which greatly assisted us in enhancing our scheme. We revised our scheme and added the "Adding New Device" phase to section 4 of the Proposed Scheme Constructions.

Reviewer 2 Report
In the manuscript, the authors have proposed a certificateless aggregate signature (CLAS) scheme based on the hyperelliptic curve cryptography (HECC) to improve the security of VANETs. The manuscript is well organized overall. However, some concerns still need to be addressed before the acceptance.
1. The mathematical expression of Eq. (1) and its related explanations need to be improved.
2. In Subsection 3.1, the definition of Delta is informal and needs to be improved.
3. Subsection 3.4 network model follows Subsection 3.1. Where are Subsection 3.2 and 3.3?
4. In the proposed scheme, there is a centralized third-party trusted authority (TA). How to ensure the security of the centralized TA? What if it is attacked and compromised? Many research efforts are working on the decentralized security and privacy preservation mechanisms in recent years.
5. Could the authors please revise Section 4 to improve the readability?
6. In Subsection 6.1, there are two Table 2s.
7. Table 4 and Figure 3 deliver exactly the same information. Is it really necessary to keep both?
8. In the analysis of communication cost, the authors assume that the bits occupied by different types of communication payloads, namely, plaintext, bilinear pairing, elliptic curve, and hyperelliptic curve. Is there any reference that can support this assumption?
9. Figure 4 delivers exactly the same information as the third column of Table 3. Is it really necessary to keep both?
Moderate editing of English language required, particularly section 4 should be revised to improve the readability.
Author Response
In the manuscript, the authors have proposed a certificateless aggregate signature (CLAS) scheme based on the hyperelliptic curve cryptography (HECC) to improve the security of VANETs. The manuscript is well organized overall. However, some concerns still need to be addressed before the acceptance.
Response: We would like to express our sincere gratitude for the valuable time and diligent effort the esteemed reviewer devoted to the evaluation of our research paper entitled ” An Efficient and Secure Certificateless Aggregate Signature Scheme for Vehicular Ad-Hoc Network”. The reviewer comments have been very helpful, and we are glad to have an opportunity to address your issues and make our work better. After carefully considering the feedback received , we have made the following revisions to the paper:
- The mathematical expression of Eq. (1) and its related explanations need to be improved.
Response: We value such suggestions. We have made the changes according to reviewer suggestions.
- In Subsection 3.1, the definition of Delta is informal and needs to be improved.
Response: We value such suggestions. We have made the changes according to reviewer suggestions.
- Subsection 3.4 network model follows Subsection 3.1. Where are Subsection 3.2 and 3.3?
Response: We have made the corrections. Thank you for highlighting it.
- In the proposed scheme, there is a centralized third-party trusted authority (TA). How to ensure the security of the centralized TA? What if it is attacked and compromised? Many research efforts are working on the decentralized security and privacy preservation mechanisms in recent years.
Response: The fact that a centralized trusted authority (TA) lacks full access to private keys ensures its security. The TA is responsible for generating a portion of the private key while the user generates the remaining portion. In the event of an attack on the TA, the attacker will be unable to access the private key in its full form. This is how TA security is maintained.
- Could the authors please revise Section 4 to improve the readability?
Response: We have enhanced section 4's readability. We have added an easy-to-understand scheme construction (see Figure 4) in order to improve scheme comprehension.
- In Subsection 6.1, there are two Table 2s.
Response: we have made the corrections and the changes has been highlighted in Subsection 6.2.
- Table 4 and Figure 3 deliver exactly the same information. Is it really necessary to keep both?
Response: We agreed with the reviewer's observations. The figures are readily interpretable.

Reviewer 3 Report
In this paper, the authors proposed a certificateless aggregate signature scheme based on the hyperelliptic curve cryptography. But supplementation and additional explanations seem necessary for the following concerns:
1.In order to better support the author's conclusion, it is recommended to introduce the experiment simulation environment and parameters.
2.Future research should be included at the end of conclusion.
3.In Section 6.1, please check Table 3. There is wrong table number.
4.In the experimental part, what is the calculation formula of Communication Cost of Computational Cost.
The sentences of the paper are fluent, and the English expression ability is good.
Author Response
In this paper, the authors proposed a certificateless aggregate signature scheme based on the hyperelliptic curve cryptography. But supplementation and additional explanations seem necessary for the following concerns:
Response: We would like to take this opportunity to extend our heartfelt appreciation to the reviewer for the significant amount of time and hard work they put into their assessment of our research paper. We are grateful to the reviewer for all of their insightful comments and recommendations.
1.In order to better support the author's conclusion, it is recommended to introduce the experiment simulation environment and parameters.
Response: The experimental setup has been provided and the modifications have been highlighted in section 6.2.
2 .Future research should be included at the end of conclusion.
Response: Future work has been included in the conclusion, and revisions have been highlighted.
3.In Section 6.1, please check Table 3. There is wrong table number.
Response: Correction has been made. Thank you for highlighting it.
4.In the experimental part, what is the calculation formula of Communication Cost of Computational Cost.
Response: The following formulas have been added to sections 6.2 and 6.3, and the modifications have been highlighted.
For Bilinear Pairing based scheme, we used the following formulas for computational cost:
Signing Cost = Number of Bilinear Pairing Operations * Time required for Single Bilinear Paring Operation
Verification Cost = Number of Bilinear Pairing Operations * Time required for Single Bilinear Paring Operation
For Elliptic Curve based scheme, we used the following formulas for computational cost:
Signing Cost = Number of Elliptic Curve Operations * Time required for Single Elliptic Curve Operation
Verification Cost = Number of Elliptic Curve Operations * Time required for Single Elliptic Curve Operation
For Hyper Elliptic Curve based scheme, we used the following formulas for computational cost:
Signing Cost = Number of Hyper Elliptic Curve Operations * Time required for Single Hyper Elliptic Curve Operation
Verification Cost = Number of Hyper Elliptic Curve Operations * Time required for Single Hyper Elliptic Curve Operation
Communicational cost formula for bilinear pairing:
|Message| + |Total number of Transmitted parameters |* 1024
Communicational cost formula for Elliptic Curve:
|Message| + |Total number of Transmitted parameters| * 160
Communicational cost formula for Hyper Elliptic Curve:
|Message| + |Total number of Transmitted parameters |* 80

Reviewer 4 Report
Thank you for your submission. I really like the number of details, but I was wondering if there are any possible examples with less math to showcase the abilities of the system - so readers with less knowledge about the topic can still understand the baseline.
Author Response
Thank you for your submission. I really like the number of details, but I was wondering if there are any possible examples with less math to showcase the abilities of the system - so readers with less knowledge about the topic can still understand the baseline.
Response: First and foremost, we are extremely grateful to the reviewer for evaluating our article. In consideration of the reviewers' comments, we have added the syntax phase (Section:3.4) so that readers with less mathematical knowledge can readily comprehend the proposed scheme.

Round 2
Reviewer 1 Report
The authors have resolved the comments of the reviewer so it is recommended for publication now. However, the current format of the paper is hard to read etc. This is an editorial issue and I hope that the publishing company can help the authors to reedit the format.
Reviewer 2 Report
I have no further comments.
The quality of writing could be further improved.
Reviewer 3 Report
The authors have made the necessary revisions and answered all the queries satisfactorily.
Minor editing of English language required.